# Influence of Perceived Maternal Self-Efficacy on Exclusive Breastfeeding Initiation and Consolidation: A Systematic Review

**DOI:** 10.3390/healthcare12232347

**Published:** 2024-11-24

**Authors:** Steven Saavedra Sanchez, Isabel Rodríguez-Gallego, Fatima Leon-Larios, Elena Andina-Diaz, Rosa Perez-Contreras, Juan D. Gonzalez-Sanz

**Affiliations:** 1Interdisciplinary Gender Studies PhD Program, University of Huelva, 21071 Huelva, Spain; stevensaavedrasanchez@gmail.com; 2Virgen del Rocío University Hospital, 41013 Seville, Spain; 3Red Cross Nursing University Centre, University of Seville, 41009 Seville, Spain; 4Nursing Department, School of Nursing, Physiotherapy and Podiatry, University of Seville, 41009 Seville, Spain; fatimaleon@us.es; 5Nursing and Physiotherapy Department, Health Sciences School, University of León, 24004 León, Spain; elena.andina@unileon.es; 6Nursing Department, University of Huelva, 21071 Huelva, Spain; rosa.perez@denf.uhu.es; 7Nursing Department, COIDESO Research Centre, University of Huelva, 21071 Huelva, Spain; juan.gonzalez@denf.uhu.es

**Keywords:** self-efficacy, self-confidence, breastfeeding, exclusive breastfeeding, systematic review

## Abstract

Background/Objectives: Different international organizations recommend exclusive breastfeeding during the neonate’s first six months of life; however, figures of around 38% are reported at the global level. One of the reasons for early abandonment is the mothers’ perception of supplying insufficient milk to their newborns. The objective of this research is to assess how mothers’ perceived level of self-efficacy during breastfeeding affects their ability to breastfeed and the rates of exclusive breastfeeding up to six months postpartum. Methods: A systematic review for the 2000–2023 period was conducted in the following databases: Cochrane, Web of Science, Scopus, PubMed, Science Direct, and CINAHL. Original articles, clinical trials, and observational studies in English and Spanish were included. Results: The results comprised 18 articles in the review (2006–2023), with an overall sample of 2004 participants. All studies were conducted with women who wanted to breastfeed, used the Breastfeeding Self-Efficacy Scale or its short version to measure postpartum self-efficacy levels, and breastfeeding rates were assessed up to 6 months postpartum. Conclusions: The present review draws on evidence suggesting that mothers’ perceived level of self-efficacy about their ability to breastfeed affects rates of exclusive breastfeeding up to 6 months postpartum. High levels of self-efficacy are positively related to the establishment and maintenance of exclusive breastfeeding; however, these rates decline markedly at 6 months postpartum

## 1. Introduction

Various international organizations support exclusive breastfeeding (EBF) during the newborn’s first six months of life, and subsequently complementing it with food options up to the age of two years or more [1,2,3].

Both for newborns and for breastfeeding (BF) mothers, the benefits of EBF are many and are widely described in the literature [4,5,6]. In addition, it can be pointed out that BF is a key determining factor in promoting public health and reducing inequalities in health [7]. Despite all the above, globally, only 38% of the newborns receive EBF during their first six months of life [4]. The BF rates decrease rapidly during the first weeks postpartum, and it is in the first month that the most noticeable change is found in the number of women who interrupt BF [8].

Demographic, physiological, and psychological factors can interfere both positively and negatively in BF interruption or maintenance [9,10,11,12,13]. Among the psychological factors is the mothers’ perception of supplying insufficient milk to their newborns. Although this condition is usually called hypogalactia, this perception is not always a true case of the condition (non-production of milk resulting from organic factors) but is often due to scarce or non-existent production related to an inadequate BF technique and to the technical aspects of BF (such as latch, position of the newborn and of the mother, etc.). This perception of BF non-efficacy is the most frequent reason for BF abandonment, which makes self-efficacy (SE) an essential factor for EBF initiation [8,14].

A large number of the studies on SE have been based on Albert Bandura’s concept [15,16,17,18]. Dennis adapted this concept to the reproductive field, defining SE regarding EBF as the mother’s self-confidence in her ability to breastfeed her newborn [19]. Since then, maternal SE has received considerable attention as a predictor of health-related behaviors [20,21,22,23] in addition to being a factor that exerts an influence on maternal satisfaction with EBF [24].

In relation to the interventions performed, several programs have been developed in an attempt to increase maternal breastfeeding self-efficacy (BSE), mainly targeted at mothers at high risk of EBF abandonment in the first weeks postpartum [11,25,26].

The aim of the review is to assess how mothers’ perceived level of self-efficacy during breastfeeding affects their ability to breastfeed and the rates of exclusive breastfeeding up to six months postpartum.

## 2. Materials and Methods

A systematic literature review was conducted according to the guidelines established by the 2020 Preferred Reporting Items for Systematic Reviews (PRISMA) (Appendix A) [27], using qualitative literature synthesis. A detailed protocol outlining the methodology and search strategies used in this review has been published on Preprints.org (DOI: https://doi.org/10.20944/preprints202411.1098.v1)

Table 1 shows the PICO criteria (participants, intervention, comparison, and outcomes) used for the inclusion of studies.

### 2.1. Search Strategy

A search of the literature from January 2000 to December 2023 was conducted. The systematic literature search was conducted in the following electronic databases: Cochrane, Web of Science (WoS), Scopus, PubMed, Science Direct, and the Cumulative Index to Nursing and Allied Health Literature (CINAHL).

The designed search strategy was conducted by combining the Medical Subject Headings (MeSH) and Descriptors in Health Sciences thesaurus with free terms as synonyms of the descriptors in the literature, through the use of AND and OR Boolean operators. The MeSH term employed was “self-confidence”, and those for DeCS were “self-efficacy”, “breastfeeding”, “lactation”, and “exclusive breastfeeding”. The free terms chosen were the following: “self-reliance” and “breastfeeding rate” (Table 2).

### 2.2. Inclusion/Exclusion Criteria

The inclusion criteria were as follows: (1) clinical trials or observational studies published in English and/or Spanish, (2) the subjects studied were postpartum women in whom the variable SE in their ability to breastfeed was present or, in its absence, related terms were used as synonyms, (3) SE was measured using a validated tool, (4) the influence of SE on the rate of EBF was studied, and (5) the studies met the PICO criteria described.

The studies excluded were studies not published in English or Spanish, studies carried out with puerperal women with any disease during pregnancy or puerperium (diabetes, pre-eclampsia, etc.), studies carried out with preterm newborns, studies that did not assess variable SE or EBF rates, and studies that did not meet all the PICO criteria.

### 2.3. Data Extraction

The article selection process began with the paired search conducted by two of the authors. The eligible studies retrieved from the six bibliographic databases (n = 250 records) were imported into the Mendeley^®^ bibliographic reference manager (version 2.126.0) and duplicates were removed (n = 44). The search in the gray literature did not yield relevant results. The pre-selected records (n = 206) were examined in two stages. As a first step, the titles and abstracts were evaluated considering eligibility regarding the inclusion criteria defined according to the PICO framework, eliminating n = 171 records. Secondly, the remaining full-text articles that were selected (n = 35) were thoroughly read to evaluate their inclusion in the review. Once the eligibility process was complete, another two authors assessed the methodological quality and the biases of the potentially useful studies; this improved the screening of the results to obtain more complete and relevant information, thus enhancing the quality of the study. The degree of agreement between both researchers in terms of evaluating eligibility of the studies was assessed using Kappa’s statistical test, with high agreement indicated (Kappa statistics = 0.85). The articles excluded were those that did not meet all the inclusion criteria, did not respond to the review objective, or were focused on studying any variable other than SE regarding ability to breastfeed (n = 17). Finally, the set of articles included in the current systematic review amounted to a total of 18 records. The PRISMA flow diagram is presented in Figure 1.

### 2.4. Data Analysis

The tool described by López de Argumedo et al. [28] for systematic reviews was used to assess the quality level and evaluate the risk of bias in the studies (Appendix A). This tool appraises six areas to assess the quality of the evidence contributed by each study included. Its purpose is to provide a structured and standardized way to identify limitations in the studies included in a review, in order to improve the interpretation of the findings and assess the strength of the scientific evidence. A narrative synthesis of the data was undertaken. From each study, we extracted general data, the assessment made by the authors, and key findings in reference to the objective of the review.

## 3. Results

### 3.1. Characteristics of the Included Studies

Of the total of 18 studies included, 13 (72.2%) were conducted in Asia, 4 (22.2%) in North America, and 1 in Europe (5.6%); 72.22% were experimental studies and 27.8% area observational studies.

All the studies were conducted with women who wanted to breastfeed, with sample sizes varying from 30 to 781, accounting for a total of 5771 participants in the 18 studies included. Eleven studies (61.1%) included only primiparous women while in the other seven (38.9%), the participants were primiparous and multiparous women.

The author, date of publication, study aim, sample, and design of the studies included in the review are shown in Table 3.

All the detailed and relevant information for data analysis, synthesis, and interpretation was collected, encompassing the following: bibliographic data, country, study design, tool used to assess SE, evaluation of the EBF rates, main results, and quality level of each study (Table 4 and Table 5).

### 3.2. Measuring Instruments and Interval

To assess levels of self-efficacy, all 13 studies reviewed in this study used the BSES [32,39,40,44] or BSES-SF [26,29,30,31,34,35,38,41,43]. The BSES was created by Dennis and Faux in 1999 to assess confidence in breastfeeding [45]. This self-administered tool consists of 33 items, all preceded by the phrase “I can always”, and is rated on a five-point Likert scale ranging from 1 (not at all confident) to 5 (very confident). Higher scores indicate greater levels of breastfeeding self-efficacy. In 2003, Dennis reduced the BSES from 33 to 14 items and renamed it the BSES-SF [46]. There is substantial evidence supporting the reliability and validity of this version as a global measure of breastfeeding self-efficacy. The reliability and validity of this instrument have been found satisfactory in different countries and populations [26,47,48,49,50,51,52].

The SE levels and the EBF rates were evaluated at different points during the postpartum period: early postpartum, 1 month postpartum (4 weeks), 2 months postpartum (8 weeks), 3 months postpartum (12 weeks), and 6 months postpartum [23,26,29,30,31,32,33,34,35,36,37,38,39,40,41,42,43,44].

Table 6 shows the author, date of publication, variables measured, instruments, reliability, and validity of the instruments of the studies included in the review.

### 3.3. Quality Assessment

The results of the analysis carried out according to the tool described by López de Argumedo et al. [28], in relation to the evaluation of methodological quality, are presented in Appendix A.

### 3.4. Self-Efficacy Levels Perceived by the Mothers About Their Ability to Breastfeed

The difference between the two scales is that the BSES consists of 33 items with a maximum score of 165 points, and the BSES-SF consists of 14 items with a maximum score of 70 points [45,46].

In the studies that used BSES, the scores were 105.28 points at 6 months postpartum (IG) in the study by Ansari et al. [32], 121.44 points at 6 months postpartum (IG) in Shariat et al. [39], and 141.44 points at 2 months postpartum (IG) in Yesil et al. [44].

Regarding the other studies that used BSES-SF, the early postpartum scores were as follows: 34.8 points (IG) in the study by Awano and Shimada [30], 51.6 points in Otsuka et al. [34], 55.89 points (IG) in Chan et al. [26], 46.2 points (IG) in Tseng et al. [41], 63.66 points (IG) in Vakilian et al. [40], and 43.05 points (IG) Wong and Chien [43].

At 1 month postpartum, the scores were 59 points in McQueen et al. [31], 53.38 points in Noel-Weiss et al. [29], 53.5 points in Otsuka et al. [34], 58.8 points in Wu et al. [35], and 48.1 points (IG) in the study by Tseng et al. [41].

At 2 months postpartum, the scores were 62.46 points in the study by Araban et al. [38] and 59.85 points in that by Wu et al. [35]. At 3 months postpartum, the score in the study by Tseng et al. was 49 points (IG) [41]. At 6 months postpartum, the scores were 49.9 points (IG) in the study by Awano and Shimada [30] and 46.7 points (IG) in Tseng et al. [41].

### 3.5. Self-Efficacy and Exclusive Breastfeeding Rates

The study with the largest sample included in the current review (781 women) was a clinical trial conducted by Otsuka et al. [34] in Japan. This study evaluated the early postpartum EBF rates and those at 4 weeks and at 12 weeks. Early postpartum, the EBF rate was 88% (mean of both groups). However, at 4 weeks postpartum, the EBF rates were higher in the women whose maternal self-efficacy levels increased (IG) than in the other groups; 73.4% maintained EBF in the group, with the highest score (53.5 points in BSES-SF). Despite this, the EBF rates presented a marked reduction at 12 weeks postpartum, falling to 47%, and the study argued that the intervention was therefore not effective in increasing or maintaining the SE levels at 12 weeks postpartum.

In this same line, Shariat et al. [39] evaluated EBF rates early postpartum and at 6 months postpartum, reporting that they dropped from 87.5% to 40.9% between early postpartum and 6 months postpartum in the IG, and from 75.4% to 23.5% in the CG. The difference in the percentages of BF rates lay in the SE levels: in the IG, the BSES scores increased after one month of intervention, up to 20 points more than in the CG. Despite that, the mean BSES scores at 6 months were similar (121.44 vs. 122.52). The variable that marked the difference was “anxiety” (which is not studied in this review): higher anxiety levels were related to lower SE levels, and the mothers in the IG presented lower anxiety levels. As in the aforementioned study [34], the EBF rates at 6 months were markedly reduced compared with early postpartum and the first postpartum weeks, and maternal SE levels were one of the factors that exerted a notable influence.

Unlike the previous studies, an intervention study performed by Ansari et al. [32] to improve SE showed that the increase in BSE sustained high EBF rates over time, with 73.3% at 6 months postpartum.

In the study by Awano and Shimada [30], which involved a sample consisting exclusively of primiparous women (which can exert an influence on the results because of the lack of previous BF experience, compared with other studies that included multiparous women), the maternal SE levels exerted a major impact on the EBF rates at 4 weeks postpartum. In that study, the early postpartum EBF rates were 90% in the IG and 89% in the CG; however, they remained at 90% in the IG at 1 month postpartum and dropped to 65% in the CG, with the SE levels also increasing in the IG throughout this period. The SE levels at 1 month postpartum were higher in the IG than in the CG (49.9 vs. 46.5 points on the BSES-SF, respectively).

However, in the study by Wu et al. [35], which was also conducted with primiparous women, the EBF rates were only 60% at 8 weeks postpartum in both groups, although the BSES-SF scores were similar to those found in Awano and Shimada [30]. The main cause was the mothers’ perception of supplying insufficient milk to their newborns. Nevertheless, higher SE levels were associated with EBF maintenance.

Other factors, such as maternal educational level or the mothers’ previous BF experience, are not addressed in the current review; however, it is necessary to take them into account in future studies, as they also exert an influence on EBF rates.

Two of the most recent studies included—Tseng et al. [41] and Vakilian et al. [40]—conducted randomized controlled trials with primiparous mothers. Vakilian et al. [40] evaluated EBF rates only at 1 month postpartum, obtaining 89.2% in the IG (63.66 points on the BSES-SF) and only 55.5% in the CG (57.04 points on the BSES-SF). The positive impact of the interventions on maternal SE levels during the first weekz postpartum is consistent with the results found by Otsuka et al. [34] and by Shariat et al. [39]. In these studies, the same results were not obtained in the subsequent weeks: although the mothers’ SE levels were maintained, the rates were markedly reduced.

This was also the case in the study by Tseng et al. [41]. After the intervention, the study determined the EBF rates at four time points: early postpartum and at 1, 3, and 6 months postpartum. The EBF rates were higher in the IG compared with the CG, although they varied in both groups at the different time points studied. The differences in the EBF rates were reflected in the BSES-SF scores, with a mean of 48 points in the IG versus 40 points in the CG. Another study, by Chan et al. [26], where the EBF rates were assessed early postpartum and at 1, 2, and 6 months postpartum, showed similar results to those of the previous study. The EBF rates presented a marked reduction at 6 months postpartum: 40%, 37.2%, 31.4%, and 11.4%, respectively, in the IG with a mean score of 50 points on the BSES-SF, and 22.2%, 13.9%, 5.5%, and 5.6% in the CG with a mean score of 40 on the BSES-SF.

In the study conducted by Wu et al. [23], only 25% of the participants reported BF during their postpartum hospitalization, a percentage much lower than that reported in previous studies, although the study also established a significant relationship between SE and EBF rates. In the oldest study, by Noel-Weiss et al. [29], EBF rates at 8 weeks were maintained at 64%, which correlated with the increase in SE levels in both groups.

Finally, the two most recent studies were by Wong and Chien [43] and Yesil et al. [44] (2023). In the study by Yesil et al. [44], there were significant differences in EBF rates, with 72.5% in the IG providing EBF at birth compared with only 30% in the CG.

In a study during the COVID-19 pandemic in China [43], where SE scores were low (43.05 points on the BSES-SF), only 53–54% of mothers gave EBF at 2 months postpartum.

## 4. Discussion

This systematic review assessed how mothers’ perceived levels of self-efficacy during breastfeeding affect their ability to breastfeed and their rates of EBF up to six months postpartum. It is necessary to take into account the demographic and cultural characteristics of mothers that may influence breastfeeding, whether they are primiparous or multiparous, and whether or not they have previous breastfeeding experience. It should also be remembered that exceptional situations such as the COVID-19 pandemic may create additional difficulties in promoting BF.

The instruments used to perform the SE measurements in the articles included in the current review [26,29,30,31,32,34,35,38,39,40,41,42,43] are among the most frequently employed in the international scientific community, showing homogeneity in the conceptualization of development, content, construction, and predictive validity [53]. It is indeed a reality that certain disparities were observed in the measurement intervals of the studies reported, which may hinder interpretation of how SE evolved during the postpartum period.

The SE levels reported in the articles reviewed [26,29,30,31,32,34,35,38,39,40,41,42,43,44] showed scores in line with what has been described in the literature in terms of all their measurements; we can compare them to the study conducted by Degrange et al. [20] where the threshold score for BSES was defined at 116/165, which would be equivalent to 49/70 in BSES-SF.

Several publications [26,30,32,34,35,38,39,40,41,42] have used two groups to study the impact of maternal SE on the ability to maintain EBF. The results showed that pregnant women with less SE in their ability to breastfeed presented significantly more chances of interrupting it [34,39,40,41,42].

Ansari et al. [32] reported a significant relationship between SE and the duration of EBF. SE is a modifiable variable that can be improved through the implementation of appropriate programs, but factors such as gestational age and maternal education level must be taken into account. Another intervention conducted in Iran [38] agreed with Ansari et al. [32] that interventions that help increase SE levels can improve EBF rates.

In the intervention conducted by Wu et al. [35] in China, EBF rates were below 60% at 4 and 8 weeks postpartum in both groups, despite having SE scores similar to those in other studies such as Awano and Shimada [30]. The main reasons were maternal perception of insufficient milk supply, lack of family support, and the limited knowledge of professionals regarding breastfeeding. Regardless of the interventions’ effectiveness, these cultural considerations influence SE levels in China and, consequently, EBF outcomes. This may be a differentiating factor compared with results from other studies such as Awano and Shimada [30], conducted in a different culture. Cultural aspects of BF also influenced the results of interventions in primiparous women in the studies by McQueen et al. [31] and Noel-Weiss et al. [29]. Nearly half of the mothers intended to breastfeed beforehand, with BF rates remaining around 60–70% at 8 weeks postpartum in both studies. This was correlated with increasing SE levels as the weeks passed.

Otsuka et al. [34] found a positive correlation between maternal SE and EBF rates in hospitals adhering to United Nations and WHO breastfeeding guidelines, compared with those that did not. Women attending hospitals supporting these guidelines had higher SE scores and EBF rates. Increased maternal SE levels generally led to higher EBF rates and duration, though prior BF experience also played a role. The study included both first-time and experienced mothers but did not specifically analyze how prior experience affected the outcomes.

It is crucial to consider cultural variations when analyzing the impact of maternal self-efficacy on breastfeeding, as different cultural contexts can significantly influence beliefs, practices, and the support mothers receive regarding breastfeeding. For example, studies conducted in Asia have shown that educational interventions based on self-efficacy theory have a particularly strong impact on maternal self-efficacy and exclusive breastfeeding rates compared with other cultural contexts, due to the central role of community norms and family support in these societies [54]. These cultural differences not only impact how interventions are implemented but also how mothers interpret and act upon the information received. Additionally, it is important to highlight the distinction between clinical and statistical significance in these studies. While the results show a statistically significant increase in maternal self-efficacy and breastfeeding duration, it is also necessary to interpret the clinical relevance of these changes—specifically, how these results translate into tangible long-term health benefits for both mothers and infants. Including a broader discussion of clinical significance could better contextualize the findings across different cultures and populations.

The most recent studies including primiparous mothers [41,42] showed that EBF rates positively correlated with SE scores. In the most recent study [44], which included both primiparous and multiparous mothers, previous BF experience influenced SE levels and EBF rates. The intervention was effective in boosting SE levels, maintaining EBF rates up to 12 weeks postpartum. An interesting study because it was developed in the middle of the COVID-19 pandemic in China [43], the SE scores were low and only half of the mothers gave EBF at 2 months postpartum. The pandemic had negative influences on SE levels and EBF rates because direct contact with mothers was not possible.

Unlike other factors, SE is potentially modifiable with interventions conducted by health professionals [55,56]. Its role as one of the factors positively associated with EBF initiation and duration is acknowledged [21,22] even in premature newborns [25]. In addition, SE has shown an additional positive impact on EBF rates from early postpartum to 6 months postpartum [20,32,36,39,40,55] and has also been identified as a significantly relevant factor for BF in future pregnancies [20].

Meanwhile, maternal education is a key factor in SE and EBF rates, as several studies suggest. Women with higher educational levels tend to exhibit greater perceived self-efficacy, which translates into a higher ability to initiate and maintain exclusive breastfeeding [54]. Additionally, education level may influence how breastfeeding information is interpreted, the capacity to seek support, and decision making during the postpartum period [57,58]. In this sense, formal education provides mothers with cognitive and critical tools that allow them to better handle breastfeeding challenges, facilitating the adoption of healthy practice [54]. However, education is not limited solely to academic training; specific educational interventions on breastfeeding, led by healthcare professionals, play a crucial role in increasing maternal SE, especially in women with fewer formal educational resources [59]. This highlights the need to design and implement inclusive and accessible educational programs for all mothers, addressing their particular needs and promoting successful breastfeeding, regardless of their educational backgrounds [54]. Although the studies reviewed do not deeply analyze this aspect, the data suggest that educational interventions may be essential for improving breastfeeding outcomes in more vulnerable populations [54].

Despite the impact of the interventions on the levels of SE and the rates of EBF at 6 months postpartum, the figures achieved are still insufficient with respect to the WHO [4] criteria, so it is necessary to continue investigating how to increase these rates.

Furthermore, it is the generalization of the included studies (external validity) that the results obtained cannot be extrapolated to the general population because the studies did not cover sufficiently heterogeneous populations (most of the samples come from hospitals in a single city).

## 5. Conclusions

The SE level in relation to mothers’ ability to breastfeed affects EBF rates up to 6 months postpartum. Consequently, higher SE levels are positively associated with EBF initiation and maintenance. EBF rates are maintained in early postpartum and at 1 month after birth when the breastfeeding SE levels are high. However, despite interventions’ positive impact on EBF rates at 6 months postpartum, even maintaining high SE levels, these rates are markedly reduced during this period and are insufficient in relation to the objectives proposed by the WHO.

Perceived SE in relation to the ability to breastfeed is a modifiable factor, so it is pertinent to identify early mothers who lack this self-confidence in their breastfeeding ability and to implement effective interventions to increase maternal SE levels.

Given the importance of maternal SE in EBF, it is recommended that future research focuses on the development and evaluation of specific interventions aimed at increasing SE levels in BF mothers. These interventions could include training programs that provide information on BF techniques, as well as support groups where mothers can share experiences and receive positive feedback. Additionally, it will be valuable to implement longitudinal studies to assess the impact of these interventions on SE levels and EBF rates, thereby allowing the identification of effective strategies and their adaptation according to the individual needs of mothers. In this way, we can contribute to improving BF outcomes and supporting mothers on their journey toward a successful BF experience.

## Figures and Tables

**Figure 1 healthcare-12-02347-f001:**
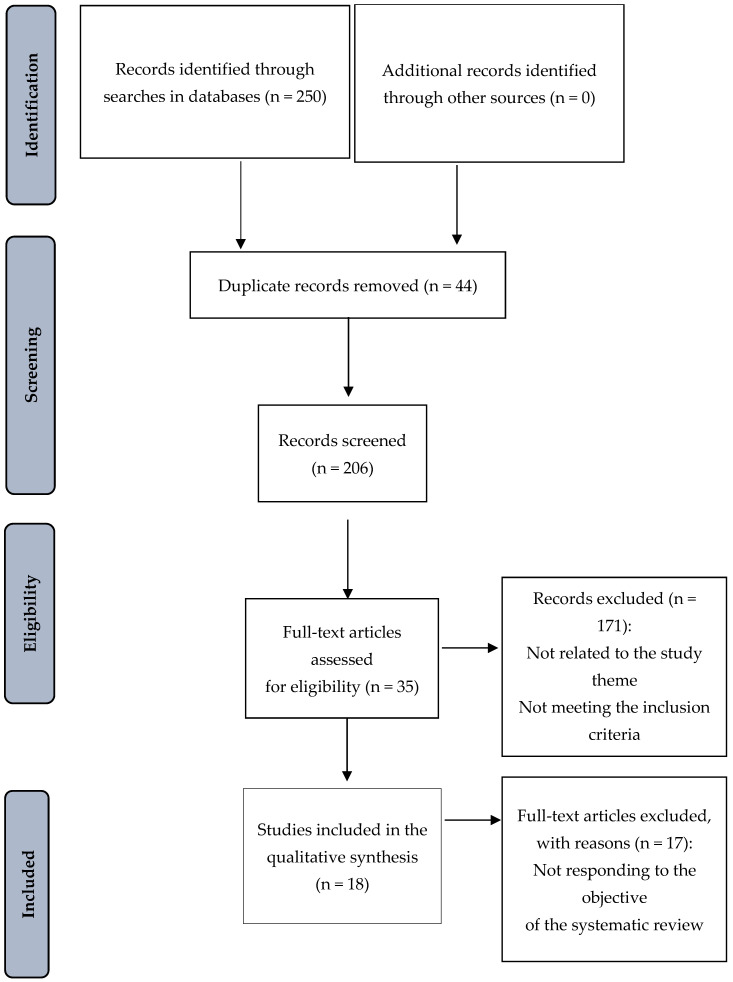
Flow diagram corresponding to the selection of articles according to PRISMA.

**Table 1 healthcare-12-02347-t001:** PICO Criteria.

Parameters Criteria	
Participants	Puerperal women who wanted to breastfeed.
Intervention	Individual or group intervention or program carried out to promote and support BSES and BF rates. Also, monitoring of breastfeeding progress.
Comparison	Puerperal women with high and low levels of SE, respectively. Usual care.
Outcome	EBF rates up to 6 months postpartum.

The PICO format was used to prepare the research question.

**Table 2 healthcare-12-02347-t002:** Search strategy in the databases.

Database	Strategy	Results
CINAHL	((TITLE “self efficacy”) AND (TITLE (breastfeeding OR lactation OR “exclusive breastfeeding”)) AND (ABSTRACT “breastfeeding rate”)) ((ABSTRACT (“self confidence” OR “self reliance”)) AND (ABSTRACT ((breastfeeding OR lactation OR “exclusive breastfeeding”)) AND (ABSTRACT “breastfeeding rate”))	11
Cochrane	“self efficacy” in Title AND (breastfeeding OR lactation OR “exclusive breastfeeding”) in Title Summary Keyword AND “breastfeeding rate” in Title Summary Keyword	15
PUBMED	(“self efficacy” [Title]) AND ((breastfeeding OR lactation OR “exclusive breastfeeding”)) AND (“breastfeeding rate”)	12
Science Direct	self efficacy” (Tittle) AND (breastfeeding OR lactation OR “exclusive breastfeeding”) AND ”breastfeeding rate” (“self confidence” OR “self reliance”) AND (breastfeeding OR lactation OR “exclusive breastfeeding”) AND ”breastfeeding rate	127
SCOPUS	(TITLE (“self efficacy”) AND TITLE ((breastfeeding OR lactation OR “exclusive breastfeeding”)) AND TITLE-ABS-KEY (“breastfeeding rate”)) (TITLE ((“self confidence” OR “self reliance”)) AND TITLE ((breastfeeding OR lactation OR “exclusive breastfeeding”)) AND TITLE-ABS-KEY (“breastfeeding rate”))	37
Web of Science	self efficacy” (Topic) AND (breastfeeding OR lactation OR “exclusive breastfeeding”) (Topic) AND ”breastfeeding rate” (Topic) (“self confidence” OR “self reliance”) (Topic) AND (breastfeeding OR lactation OR “exclusive breastfeeding”) (Topic) AND ”breastfeeding rate” (Topic)	48

**Table 3 healthcare-12-02347-t003:** Author, date of publication, study aim, sample, and design of the studies included in the review.

Author, (Year)	Objective	Design	Participants	Sample Size
Noel-Weiss, J. et al. (2006) [29]	To determine the effects of a prenatal breastfeeding workshop on maternal BSE and BF duration.	Randomized controlled trial	Primparous women	110
Awano, M & Shimada, K. (2010) [30]	To develop a self-care programme for BF aimed at increasing mothers’ breastfeeding confidence and to evaluate its effectiveness.	Quasi-experimental	Primiparous women	117
McQueen, K.A. et al. (2011) [31]	To pilot test a newly developed BSE intervention.	Randomized controlled trial	Primiparous women	149
Ansari, S. et al. (2014) [32]	To determine the effect of an educational programme on BSE and the duration of EBF in pregnant women.	Randomized controlled trial	Primiparous women	120
Glassman, M.E. et al. (2014) [33]	To quantify early changes in amounts of BF and to explore the role of BSE and sociocultural factors associated with any BF and EBF in the first 4–6 weeks postpartum.	Observational and descriptive	Primiparous and multiparous women	209
Otsuka, K. et al. (2014) [34]	To evaluate the effect of an SE intervention on BSE and EBF.	Clinical trial	Primiparous and multiparous women	781
Wu, D.S. et al. (2014) [35]	To evaluate the effects of a breastfeeding intervention on primiparous mothers’ BSE, BF duration, and exclusivity at 4 and 8 weeks postpartum.	Randomized clinical trial	Primiparous women	74
Henshaw, E.J. et al. (2015) [36]	To evaluate the relationships among BSE, mood, and breastfeeding outcomes in primiparous women. A secondary purpose was to explore self-reported reasons for difficult emotional adjustment during the transition to motherhood.	Prospective study	Primiparous women	146
Chan, M.Y. et al. (2016) [26]	To investigate the effectiveness of a self-efficacy-based breast feeding educational programme (SEBEP) in enhancing BSE, BF duration, and EBF rates.	Clinical trial	Primiparous women	71
Ip, W.Y. et al. (2016) [37]	To examine the relative effect of maternal BSE and selected relevant factors on the EBF rate at 6 months postpartum.	Cohort study	Primiparous and multiparous women	562
Araban, M. et al. (2018) [38]	To determine the effects of a prenatal BSE intervention on BSE and BF outcomes.	Randomized controlled trial	Primiparous women	120
Shariat, M. et al. (2018) [39]	To examine the effect of interventions leading to increased awareness, knowledge, and SE regarding EBF and duration of BF.	Randomized clinical trial	Primiparous and multiparous women	129
De Roza, J.G. et al. (2019) [40]	To examine the factors that affect EBF.	Observational and descriptive study	Primiparous and multiparous women	400
Tseng, J.F. et al. (2020) [41]	To develop an integrated BF education programme based on SE theory, and evaluate the effect of the intervention on first-time mothers’ BSE and attitudes.	Randomized clinical trial	Primiparous women	93
Vakilian, K. et al. (2020) [42]	To evaluate the effects of home-based education intervention on exclusivity and promoting the rates of BSE.	Randomized clinical trial	Primiparous women	130
Wu, S.F.V. et al. (2021) [23]	To assess women’s intention to breastfeed and knowledge and SE regarding BF following childbirth, and to identify the factors associated with postpartum breastfeeding during women’s hospital stays.	Descriptive and longitudinal study with pre-/post-test	Primiparous and multiparous women	120
Wong, M.S. & Chien, W.T. (2023) [43]	To examine the effects of different approaches to educational and supportive interventions that can help sustain BF and improve BSE for primiparous postnatal women, and to identify key characteristics of the effective interventions in terms of delivery time, format, and mode, main components, use of theoretical framework, and number of sessions.	Randomized clinical trial	Primiparous women	30
Yesil, Y. et al. (2023) [44]	To examine the effect of hospital-based group BF education provided to mothers before discharge from hospital on mothers’ SE and on increasing BF rates.	Randomized clinical trial	Primiparous and multiparous women	80

Note: BSE = breastfeeding self-efficacy; BF = breastfeeding; EBF = exclusive breastfeeding; SE = self-efficacy.

**Table 4 healthcare-12-02347-t004:** Characteristics of the articles included in this review (experimental studies).

Author, (Year)	Country	Design	Tool to Assess SE	Assessment of the EBF Rates	Main Results	Quality
Noel-Weiss, J. et al. (2006) [29]	Canada	Randomized controlled trial	BSES-SF	4 and 8 weeks postpartum	SE scores increased in both groups at 4 and 8 weeks. These SE scores positively correlated with the maintenance of EBF, with the mean EBF rate of both groups being 68% at 8 weeks.	Medium
Awano, M & Shimada, K. (2010) [30]	Japan	Quasi-experimental	BSES-SF	Early postpartum and 4 weeks postpartum	The BSES-SF score in the IG increased significantly from 3.8 to 49.9 one month after birth (*p* < 0.01), unlike the CG (*p* = 0.03). The early postpartum BF rate was similar in both groups; however, at 4 weeks postpartum, the EBF rate was significantly reduced to 65% in the CG compared with 90% in the IG (*p* = 0.02).	Medium
McQueen, K.A. et al. (2011) [31]	Canada	Randomized controlled trial	BSES-SF	4 and 8 weeks postpartum	Scores for SE were high in both the IG (59 points) and the CG (54.9 points). This had an impact on EBF rates, keeping them above 65% in both groups at 8 weeks postpartum. Additionally, the mothers’ prior intention to breastfeed influenced the results.	High
Ansari, S. et al. (2014) [32]	Iran	Randomized controlled trial	BSES	6 months postpartum	SE increased significantly in the IG compared with the CG 1 month after birth (*p* < 0.001). EBF duration was significantly longer in the IG (*p* < 0.001). There was a significant relationship between SE and EBF duration (*p* < 0.001).	Medium
Otsuka, K. et al. (2014) [34]	Japan	Clinical trial	BSES-SF	Early postpartum, and 4 and 12 weeks postpartum	In the IG, there were improvements both in SE up to 4 weeks postpartum (*p* = 0.037) and in the EBF rate at 4 weeks postpartum (OR_adj_ = 2.32, 95% CI = 1.01–5.33), unlike the CG. Higher scores on the BSES-SF—and therefore, higher SE levels—were related to better results in the EBF rates.	Low
Wu, D.S. et al. (2014) [35]	China	Randomized clinical trial	BSES-SF	4 and 8 weeks postpartum	The IG obtained significantly higher SE scores and better EBF rates than the CG (*p* < 0.01) at 4 and 8 weeks. The women with higher SE levels were more prone to the EBF practice at 4 and 8 weeks postpartum (*p* < 0.01). Differences in BF duration were found at 8 weeks (*p* = 0.047), though not at 4 weeks (*p* = 0.11).	
Chan, M.Y. et al. (2016) [26]	China	Clinical trial	BSES-SF	2, 4, and 8 weeks and 6 months postpartum	SE exerted an influence on the EBF rates, which were higher in the IG than in the CG at 2 weeks (*p* < 0.01). There were no significant differences between the groups for BF duration at 6 months (*p* = 0.07)	Medium
Araban, M. et al. (2018) [38]	Iran	Randomized controlled trial	BSES-SF	8 weeks postpartum	EBF rates and self-efficacy scores were higher in the IG than in the CG at 8 weeks postpartum. There is clear evidence that increasing SE levels improves EBF rates.	High
Shariat, M. et al. (2018) [39]	Iran	Randomized clinical trial	BSES	Early postpartum, and 6, 12, 18, and 24 months postpartum	Although there were no significant differences in the BSES scores between the groups (*p* = 0.09), SE exerted a positive and significant effect on EBF duration, which was significantly longer in the IG than in the CG at 6 months (*p* < 0.01). The higher the SE levels, the more the EBF was extended.	High
Tseng, J.F. et al. (2020) [41]	Taiwan	Randomized clinical trial	BSES-SF	1 week, and 1, 3, and 6 months postpartum	The EBF rates were higher in all the IG participants, where the BSES-SF scores were also significantly better than in the CG at 1 week, 1 month, and 3 months postpartum (*p* < 0.01), with a positive relationship between SE levels and EBF duration. There were no significant differences at 6 months postpartum.	High
Vakilian, K. et al. (2020) [42]	Iran	Randomized clinical trial	BSES-SF	Early postpartum, and 1 month postpartum	There were no differences between the groups regarding the SE level early postpartum. However, the BSES-SF scores in the IG were higher after 1 month postpartum (*p* = 0.01), as was the EBF rate (*p* = 0.01).	High
Wong, M.S. & Chien, W.T. (2023) [43]	China	Randomized clinical trial	BSES-SF	2 months postpartum	Only 50% of the mothers in both groups; EBF at 2 months postpartum. BSE scores were low in both groups (43.2 IG; 42.9 CG), which may have influenced the EBF rates. Additionally, the COVID-19 pandemic also had an impact.	High
Yesil, Y. et al. (2023) [44]	Turkey	Randomized Clinical trial	BSES	Early postpartum, and 4 and 12 weeks postpartum	EBF rates were higher in the IG at birth compared with the CG (70% vs. 30%). EBF rates were maintained in the IG but not in the CG.	High

Note: CG = control group; IG = intervention group; BSES = Breastfeeding Self-Efficacy Scale; BSES-SF = Breastfeeding Self-Efficacy Scale—Short Form; BF = breastfeeding; EBF = exclusive breastfeeding; AOR = adjusted odds ratio.

**Table 5 healthcare-12-02347-t005:** Characteristics of the articles included in this review (observational studies).

Author, (Year)	Country	Design	Tool to Assess SE	Assessment of the EBF Rates	Main Results	Quality
Glassman, M.E. et al. (2014) [33]	United States	Observational and descriptive	BSES-SF	4–6 weeks postpartum	Higher SE levels were associated with higher EBF rates at 4–6 weeks postpartum (OR_adj_ = 1.18 (1.05, 1.32), where SE was a factor that presented a positive association with EBF.	Medium
Henshaw, E.J. et al. (2015) [36]	United States	Prospective study	BSES-SF	Early postpartum, 6 weeks and 6 months	Women’s mood was related to the BSE levels, which in turn were associated with EBF continuity—i.e., better mood was positively related to higher SE scores and, in turn, with better success rates in EBF continuity at 6 months postpartum (*p* < 0.01).	High
Ip, W.Y. et al. (2016) [37]	China	Cohort study	BSES-SF	Early postpartum, 1, 4, and 12 weeks postpartum	The mothers showed low SE levels with only 47.3 points on the BSES-SF. As a result, EBF rates were only 24.6% at birth, while at 6 months, almost no mother was exclusively BF, with a rate of just 0.2%.	High
De Roza, J.G. et al. (2019) [40]	Singapore	Observational and descriptive study	BSES-SF	3 and 6 months	The BSES-SF scores were significantly higher in the mothers who continued EBF at 3 and 6 months compared with those who interrupted breastfeeding (*p* < 0.01).	
Wu, S.F.V. et al. (2021) [23]	Taiwan	Descriptive and longitudinal study with pre-/post-test	BSES-SF	30–34 gestational weeks Early postpartum	The mean SE score was 41.55 (SD = 12.09). Among the factors that exerted an influence on BF and EBF duration postpartum, SE presented a statistically significant difference (*p* < 0.05). SE was one of the significant characteristics among the women who chose to breastfeed during the postpartum period compared with those who did not (*p* = 0.011)	High

Note: BSES = Breastfeeding Self-Efficacy Scale; BSES-SF = Breastfeeding Self-Efficacy Scale—Short Form; BF = breastfeeding; EBF = exclusive breastfeeding; AOR = adjusted odds ratio.

**Table 6 healthcare-12-02347-t006:** Author, date of publication, variables measured, instruments, reliability, and validity of the instruments of the studies included in the review.

Author, (Year)	Variables Measured	Instruments	Reliability and Validity of Instrument
Noel-Weiss, J. et al. (2006) [29]	SE and EBF at 4 and 8 weeks postpartum	BSES-SF	Original version of BSES-SF. Cronbach’s Alpha 0.94.
Awano, M & Shimada, K. (2010) [30]	SE, BF, and EBF in early postpartum and 4 weeks postpartum	BSES-SF	Adaptation and validation BSES-SF scale to Japan. Cronbach’s Alpha 0.94
McQueen, K.A. et al. (2011) [31]	SE and EBF at 4 and 8 weeks postpartum	BSES-SF	Original version of BSES-SF. Cronbach’s Alpha 0.94.
Ansari, S. et al. (2014) [32]	SE and EBF at 6 months postpartum	BSES	Adaptation and validation BSES to Persian. Cronbach’s Alpha 0.82
Glassman, M.E. et al. (2014) [33]	SE and EBF	BSES-SF	Adaptation and validation BSES-SFto Portuguese. Cronbach’s Alpha 0.71.
Otsuka, K. et al. (2014) [34]	SE and EBF in early postpartum, and 4 and 12 weeks postpartum	BSES-SF	Adaptation and validation BSES-SF to Japanese. Cronbach’s Alpha 0.95.
Wu, D.S. et al. (2014) [35]	SE, BF, and EBF at 4 and 8 weeks postpartum	BSES-SF	Adaptation and validation BSES-SFto Chinese. Cronbach’s Alpha 0.89
Henshaw, E.J. et al. (2015) [36]	SE and EBF	BSES-SF	Adaptation and validation BSES-SF to USA population. Cronbach’s Alpha 0.92.
Chan, M.Y. et al. (2016) [26]	SE, BF, and EBF at 4 and 8 weeks and 6 months postpartum	BSES-SF	Adaptation and validation BSES-SF to Hong Kong Chinese. Cronbach’s Alpha 0.89.
Ip, W.Y. et al. (2016) [37]	SE and EBF	BSES-SF	Adaptation and validation BSES-SF to Chinese. Cronbach’s Alpha 0.89.
Araban, M. et al. (2018) [38]	SE and EBF at 8 weeks postpartum	BSES-SF	Adaptation and validation BSES-SF to Persian. Cronbach’s Alpha 0.91.
Shariat, M. et al. (2018) [39]	SE and EBF in early postpartum, and 6, 12, 18, and 24 months postpartum	BSES	Adaptation and validation of the BSES to the population of Tehran. Cronbach’s Alpha 0.82.
De Roza, J.G. et al. (2019) [40]	SE and EBF	BSES-SF	Original version of BSES-SF. Cronbach’s Alpha 0.94.
Tseng, J.F. et al. (2020) [41]	SE and EBF at 1 week, and 1, 3, and 6 months postpartum	BSES-SF	The Cronbach’s alpha reliability of the Taiwanese version of BSES-SF was 0.95. The Cronbach’s alpha for this study was 0.93.
Vakilian, K. et al. (2020) [42]	SE and EBF in early postpartum, and 1 month postpartum	BSES	Persian version of BSES scale. Cronbach’s alpha 0.89.
Wu, S.F.V. et al. (2021) [23]	SE and EBF	BSES-SF	Adaptation and validation BSES-SF to Chinese. Cronbach’s Alpha 0.89.
Wong, M.S. & Chien, W.T. (2023) [43]	SE and EBF at 2 months postpartum	BSES-SF	Hong Kong Chinese version of the BSES-SF. Cronbach’s Alpha 0.95.
Yesil, Y. et al. (2023) [44]	SE and EBF in early postpartum, and 4 and 12 weeks postpartum	BSES	Turkish adaptation and validation of BSES. Cronbach’s Alpha 0.91

Note: BSES = Breastfeeding Self-Efficacy Scale; BSES-SF = Breastfeeding Self-Efficacy Scale—Short Form; BF = breastfeeding; EBF = exclusive breastfeeding.

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
