# Peer review of "Influence of Perceived Maternal Self-Efficacy on Exclusive Breastfeeding Initiation and Consolidation: A Systematic Review"

_healthcare, 2024, doi:10.3390/healthcare12232347_

Round 1
Reviewer 1 Report
Comments and Suggestions for Authors
Thank you for giving me the opportunity of reviewing the manuscript “Influence of perceived maternal self-efficacy on exclusive 2 breastfeeding initiation and consolidation: A systematic review” submitted to Healthcare. The authors aimed to examine how mothers' perceived level of self-efficacy during breastfeeding affects their ability to breastfeed. This systematic review included 18 articles from 2006- 2023; the overall sample size was 2004.
The paper discusses an important research question. However, there are some comments that need to be addressed before the paper can be accepted for publication.
1. One of the main concerns is that the inclusion criteria of the review mention that “only clinical trials published in English and/or Spanish” will be included. However, both clinical trials and observational studies were analyzed in this systematic review.
2. The manuscript would benefit from a thorough review of the English language, as the text is often difficult to follow and understand.
3. Many abbreviations are used in the paper, some are not necessary (e.g., NB), others need to be defined (e.g., BSES in the text)
4. The information presented in tables 3 and 4 overlap (e.g., sample size and primiparous/multiparous women). Please consider avoiding the repetition of the results.
5. The included study types/designs need to be mentioned in the Abstract.
6. The effect of maternal education is mentioned in the paper; however, a more detailed discussion would be needed in the Discussion section.
7. The authors state that they follow the PRISMA recommendations. The paper would benefit from the inclusion of a PRISMA checklist.
8. Line 93 repeatedly says “is measured”. Please correct it.
Comments on the Quality of English LanguageThe manuscript would benefit from an extensive editing of English language.
Author Response
Dear Editor and Reviewer(s):
Thank you very much for this opportunity to improve our manuscript.
Please, see below our response to your comments. You can see all the changes made in highlight in the new version of the manuscript.
Reviewer(s)' Comments to Author:
Reviewer: 1
Comments to the Author
- One of the main concerns is that the inclusion criteria of the review mention that “only clinical trials published in English and/or Spanish” will be included. However, both clinical trials and observational studies were analyzed in this systematic review.
Response: Thank you very much for the annotation. We have modified the explanation of the inclusion criteria. Please see line 91-92.
- The manuscript would benefit from a thorough review of the English language, as the text is often difficult to follow and understand.
Response: Thank you so much for your suggestion. We have now carefully proofread these sections and also asked a professional language copyeditor to copyedit the article. We hope it now reads more clearly.
- Many abbreviations are used in the paper, some are not necessary (e.g., NB), others need to be defined (e.g., BSES in the text)
Response: We have modified and explained the abbreviations for a better understanding of the text. We have retained only those abbreviations that are recognized and directly related to the study objective: breastfeeding (BF), exclusive breastfeeding (EBF), self-efficacy (SE) Breastfeeding Self-Efficacy Scale (BSES) and Breastfeeding Self-Efficacy Scale short form (BSES-SF). Other necessary abbreviations in the tables are properly explained in the table footnotes.
- The information presented in tables 3 and 4 overlap (e.g., simple size and primiparous/multiparous women). Please consider avoiding the repetition of the results.
Response: Thank you very much for your comment. We have removed repeated items. In Tables 4 and 5, the columns for participants and sample size have been removed. This information is provided for all the studies included in Table 3.
- The included study types/designs need to be mentioned in the Abstract.
Response: We have mentioned the included study types/designs in the Abstract. Please see lines 19-20.
- The effect of maternal education is mentioned in the paper; however, a more detailed discussion would be needed in the Discussion section.
Response: Thank you for your appreciation. The influence of maternal education has been detailed in the discussion section. Please review lines 379-394.
- The authors state that they follow the PRISMA recommendations. The paper would benefit from the inclusion of a PRISMA checklist.
Response: Thank you for this contribution. The PRISMA checklist has been included in Supplementary Table S1.
- Line 93 repeatedly says “is measured”. Please correct it. The manuscript would benefit from an extensive editing of English language.
Response: We have corrected line 94 with your indication. We have also carried out a complete revision of the language of the text.
Reviewer 2 Report
Comments and Suggestions for Authors
SUGGESTION FOR CHANGES IN THE FOLLOWING SECTIONS
MATERIALS AND METHODS
I suggest that the item "Intervention" in the PICO framework be clearly described, including the specific interventions that were investigated in each study included in the review. This not only improves the transparency of your research but also facilitates replication by other researchers and strengthens the scientific validity of the work.
The methodology follows PRISMA standards, but I suggest adding more details about the bias assessment in the included studies. Although the López de Argumedo tool is mentioned, the explanation of the types of biases evaluated could be more detailed. The inclusion criteria were well-defined, focusing on studies that assessed self-efficacy using validated tools (BSES or BSES-SF) and examined exclusive breastfeeding outcomes. However, the exclusion of studies with participants who had pregnancy complications (such as diabetes or preeclampsia) may have reduced the generalizability of the results, given that these specific conditions can affect self-efficacy and breastfeeding rates. Line no. 92: the word "measured" is repeated.
DISCUSSION
There should be a more in-depth analysis of cultural variations among the studies. The text mentions cultural differences, but specific details are missing on how these variables impact self-efficacy and breastfeeding. The interpretation of the results in the article is appropriate, but it could be strengthened with a more detailed discussion of clinical significance versus statistical significance.
CONCLUSIONS
The conclusion aligns with the findings and highlights that maternal self-efficacy is an important and modifiable factor in exclusive breastfeeding. It would be important to include a more explicit recommendation for future research and how specific interventions could be tested to improve self-efficacy levels.
Other considerations:
The article alternates between the use of "SE" and "self-efficacy"; "BF" and "breastfeeding"; and "EBF" and "exclusive breastfeeding." The acronyms "SE," "BF," and "EBF" should be defined at first mention and used consistently throughout the article. Some sentences are long and complex and could be simplified to improve readability.
Author Response
Dear Editor and Reviewer(s):
Thank you very much for this opportunity to improve our manuscript.
Please, see below our response to your comments. You can see all the changes made in highlight in the new version of the manuscript
I suggest that the item "Intervention" in the PICO framework be clearly described, including the specific interventions that were investigated in each study included in the review. This not only improves the transparency of your research but also facilitates replication by other researchers and strengthens the scientific validity of the work. The methodology follows PRISMA standards, but I suggest adding more details about the bias assessment in the included studies. Although the López de Argumedo tool is mentioned, the explanation of the types of biases evaluated could be more detailed.
Response: The intervention component of the PICO question has been clarified. Please review Table 1.
Excellent observation. We have provided a more detailed explanation of López de Argumedo tool. Please see lines 128-130. In addition, the detailed analysis based on this tool is specified in the supplementary table S3.
The inclusion criteria were well-defined, focusing on studies that assessed self-efficacy using validated tools (BSES or BSES-SF) and examined exclusive breastfeeding outcomes. However, the exclusion of studies with participants who had pregnancy complications (such as diabetes or preeclampsia) may have reduced the generalizability of the results, given that these specific conditions can affect self-efficacy and breastfeeding rates. Line no. 92: the word "measured" is repeated.
Response: Thank you for your comment. Although we agree with you that this approach to the study may reduce its generalizability, we thought it would be useful to differentiate between women who breastfeed after a pregnancy with and without complications, since we understand that these complications are very different and may affect self-efficacy in different ways, so it would be appropriate to study them separately and by groups (for example, those related to endocrine problems such as DM, compared to those related to mental problems such as postpartum depression, or cardiovascular problems such as HTN).
We have eliminated the repetition, please see line 94.
DISCUSSION
There should be a more in-depth analysis of cultural variations among the studies. The text mentions cultural differences, but specific details are missing on how these variables impact self-efficacy and breastfeeding. The interpretation of the results in the article is appropriate, but it could be strengthened with a more detailed discussion of clinical significance versus statistical significance.
Response: Thank you for your appreciation. It has been included information about it in the discussion, please review lines 350-364.
CONCLUSIONS
The conclusion aligns with the findings and highlights that maternal self-efficacy is an important and modifiable factor in exclusive breastfeeding. It would be important to include a more explicit recommendation for future research and how specific interventions could be tested to improve self-efficacy levels.
Response: Your contribution has been taken into account. A new paragraph has been included that makes a more explicit recommendation for future research; please review lines 414-422.
Other considerations:
The article alternates between the use of "SE" and "self-efficacy"; "BF" and "breastfeeding"; and "EBF" and "exclusive breastfeeding." The acronyms "SE," "BF," and "EBF" should be defined at first mention and used consistently throughout the article. Some sentences are long and complex and could be simplified to improve readability.
Response: We have defined the different acronyms in the text for a better understanding of the text. We have also conducted a review of the reading of the text in general.
Reviewer 3 Report
Comments and Suggestions for Authors
Dear Authors,
Your manuscript, which evaluates how the maternal perceived level of self-efficacy during breastfeeding affects the ability to breastfeed and the rates of exclusive breastfeeding, is both interesting and important. However, I would like to raise the following issues:
-
What was the rationale for including the Spanish language in the list of languages used for the article search? What value did it add? Don’t you think this could have introduced bias, as other major world languages were not included in the search strategy?
-
In relation to review biases, the authors did not address them or discuss the strategies used for their mitigation.
-
Did you register this review in PROSPERO or any other databases prior to the literature search? If so, please provide the registration number.
-
Lastly, the way the research hypothesis was formulated (to evaluate the rates of exclusive breastfeeding) and the nature of the eligible studies identified suggest that a meta-analysis could be performed, which was not conducted. For example, a meta-analysis of studies listed under the subsection "3.5. Self-efficacy and exclusive breastfeeding rates" is feasible. The authors should consider this possibility.
Author Response
Dear Editor and Reviewer(s):
Thank you very much for this opportunity to improve our manuscript.
Please, see below our response to your comments. You can see all the changes made in highlight in the new version of the manuscript.
Comments to the Author
- What was the rationale for including the Spanish language in the list of languages used for the article search? What value did it add? Don’t you think this could have introduced bias, as other major world languages were not included in the search strategy?
Response: Thank you for your comment. We have eliminated this mention since no studies in this language were identified that were finally included in the review.
- In relation to review biases, the authors did not address them or discuss the strategies used for their mitigation.
Response: It is true that studies were not eliminated from the review due to review biases; however, in the peer review conducted according to the inclusion and exclusion criteria, the degree of agreement between both researchers in terms of evaluating the eligibility of the studies was assessed using Kappa's statistical test, resulting in high agreement (Kappa statistic = 0.85).
- 3. Did you register this review in PROSPERO or any other databases prior to the literature search? If so, please provide the registration number.
Response: The registration in PROSPERO has not been completed, as it was not done from the beginning. Previously, they accepted registrations provided the reviewers had not completed their data extraction but, from 1 October 2019, PROSPERO only accepts reviews provided that data extraction has not yet started.
- Lastly, the way the research hypothesis was formulated (to evaluate the rates of exclusive breastfeeding) and the nature of the eligible studies identified suggest that a meta-analysis could be performed, which was not conducted. For example, a meta-analysis of studies listed under the subsection "3.5. Self-efficacy and exclusive breastfeeding rates" is feasible. The authors should consider this possibility.
Response: Thank you for your contribution. We have carefully considered your suggestion. The aim of the review is to assess how mothers' perceived level of self-efficacy during breastfeeding affects their ability to breastfeed and the rates of exclusive breastfeeding up to six months postpartum, regardless of whether a breastfeeding support or promotion intervention has been carried out. For this reason, a meta-analysis of the results has not been provided. Furthermore, the heterogeneity of the interventions carried out in the included clinical trials—in terms of modality, frequency, and reported outcomes—has made it impossible to subgroup the studies and produce a quantitative synthesis of the results.
Round 2
Reviewer 3 Report
Comments and Suggestions for Authors
Well done!
Author Response
Thank you very much for the time you dedicated to this manuscript and for your contributions. They have helped to improve the quality of the article.